# Shikonin Induces ROS-Dependent Apoptosis Via Mitochondria Depolarization and ER Stress in Adult T Cell Leukemia/Lymphoma

**DOI:** 10.3390/antiox12040864

**Published:** 2023-04-02

**Authors:** Piyanard Boonnate, Ryusho Kariya, Seiji Okada

**Affiliations:** Division of Hematopoiesis, Joint Research Center for Human Retrovirus Infection, Kumamoto University, Kumamoto 860-0811, Japan; m-teramoto@kumamoto-u.ac.jp (P.B.); ryushokariya@kumamoto-u.ac.jp (R.K.)

**Keywords:** Shikonin (SHK), adult T cell leukemia/lymphoma (ATLL), Reactive Oxygen Species (ROS), mitochondria depolarization, endoplasmic reticulum (ER) stress, apoptosis

## Abstract

Adult T cell leukemia/lymphoma (ATLL) is an aggressive T-cell malignancy that develops in some elderly human T-cell leukemia virus (HTVL-1) carriers. ATLL has a poor prognosis despite conventional and targeted therapies, and a new safe and efficient therapy is required. Here, we examined the anti-ATLL effect of Shikonin (SHK), a naphthoquinone derivative that has shown several anti-cancer activities. SHK induced apoptosis of ATLL cells accompanied by generation of reactive oxygen species (ROS), loss of mitochondrial membrane potential, and induction of endoplasmic reticulum (ER) stress. Treatment with a ROS scavenger, N-acetylcysteine (NAC), blocked both loss of mitochondrial membrane potential and ER stress, and prevented apoptosis of ATLL cells, indicating that ROS is an upstream trigger of SHK-induced apoptosis of ATLL cells through disruption of the mitochondrial membrane potential and ER stress. In an ATLL xenografted mouse model, SHK treatment suppressed tumor growth without significant adverse effects. These results suggest that SHK could be a potent anti-reagent against ATLL.

## 1. Introduction

Adult T cell leukemia/lymphoma (ATLL) is an aggressive T-cell malignancy related to human T-cell leukemia virus type 1 (HTLV-1) infection. Oncogenesis of ATLL is known to be regulated by two viral proteins: Transactivator protein (Tax) and HTLV-1 basic leucine zipper factor (HBZ) [1]. HTLV-1 evades the immune response by reducing Tax and activating HBZ expression. Tax plays a role in neoplastic transformation while HBZ protein is responsible for ATLL cell proliferation [2,3,4]. Tax can directly and indirectly activate the NF-kB, PI3K/AKT and JAK/stat pathways, and it contributes to initial leukemogenesis [5]. Tax also induces resistance to apoptosis by interacting with TRAF6, an inhibitor of proapoptotic Mcl-1, preventing cytochrome c release from mitochondria [6]. HBZ inhibits Bim by upregulation of Bcl-2 and FOXO3 and contributes to maintenance of ATLL [7,8,9]. Direct-targeting therapies against Tax and HBZ are being examined but have yet to succeed. Since most patients are elder, the effects of conventional chemotherapy and allogeneic hematopoietic cell transplantation are less effective [10]. Monoclonal antibodies such as mogamulizumab improve response rates, but have little effect on survival [11]. Thus, there is a need for a novel alternative drug treatment for ATLL.

Natural products are a rich source of anti-cancer reagents [12]. Shikonin (SHK) is the major chemical component of *Lithospermum erythrorhizo*, and has a broad range of pharmacological activities including anti-inflammatory, anti-oxidative stress, anti-virus, anti-bacteria, and anti-cancer effects [13,14,15]. The anti-cancer effect of SHK by induction of reactive oxygen species (ROS) has recently drawn attention [16,17,18], modulating cancer survival pathways such as PI3K/AKT/mTOR and MAPKs signaling [14,19].

In this study, we investigated the anti-ATLL activity of SHK, primarily by induction of ROS followed by loss of mitochondrial membrane potential and induction of endoplasmic reticulum (ER) stress, and finally activation of apoptosis pathway. Our findings provide an experimental basis for utilizing SHK against drug-resistant ATLL.

## 2. Materials and Methods

### 2.1. Cell Lines

Human ATLL cell lines (ED−, TL-Om1, S1T, OATL4) were kindly provided by Dr. Hidekatsu Iha (Ohita Univrsity, Ohita, Japan). HTLV-1-infected immortalized cell lines MT-2 and MT-4 were kindly provided by Dr. Kazuhiko Ide (Kumamoto University, Kumamoto, Japan). These cell lines were cultured in RPMI medium (Fujifilm Wako Pure Chemical, Osaka, Japan) containing 10% fetal bovine serum (Thermo Fisher Scientific, Waltham, MA, USA), 100 U/mL penicillin G (Meiji Seika Pharma, Tokyo, Japan) and 100 µg/mL streptomycin (Fujifilm Wako Pure Chemical, Osaka, Japan) at 37 °C with 5% CO_2_.

### 2.2. Reagents

Pan-caspase inhibitor (Q-VD-OPh, TONBO Biosciences, San Diego, CA, USA), DCFH-DA (2′-7′-dichlorodihydrofluorescein diacetate) redox-sensitive fluorescent probe (Sekisui Medical, Tokyo, Japan), JC-1 (5,5′,6,6′-tetrachloro-1,1′,3,3′-tetraethyl-benzimidazolyl-carbocyanine-iodide) (Dojindo Laboratories, Kumamoto, Japan), and Shikonin (SHK) (Sigma-Aldrich, St. Louis, MO, USA) were dissolved in DMSO. N-Acetyl-L-cysteine (NAC) (Fujifilm Wako Pure Chemical) was dissolved in RPMI medium. ISRIB (Sigma-Addrich), an integrated stress response inhibitor, was dissolved in DMSO [20].

The sources of antibodies were as follows: mAb, rabbit anti-cleaved caspase-3 (D175; #9661) mAb, rabbit anti-cleaved caspase 8 (18C8; #9496) mAb, rabbit anti-cleaved caspase 9 (D8I9E; #20750) mAb, mouse anti-CHOP (L63F7; #2895) mAb, rabbit anti-p-BCl2 (Ser70)(5H2); (#2827) mAb, rabbit anti-p-JNK (T183/Y185) (81E11; #4668) mAb, rabbit anti-JNK (56G8; #9258) mAb, rabbit anti-ATF4 (D4B8; #18815) mAb, rabbit anti-p-eIF2α (S51)(119A11; #3597) mAb, rabbit anti-cytochrome C (D18C7), horseradish peroxidase (HRP)-linked goat anti-rabbit IgG (#7074), and HRP-linked horse anti-mouse IgG (#7076) obtained from Cell Signaling Technology, Inc. (Danvers, MA, USA); mAb, mouse anti-actin (C-2) (SC-8432) mAb, mouse anti-survivin (D-8) (SC-1777) and pAbs, rabbit anti-XBP1 (M-186) (SC-7160) were obtained from Santa Cruz Biotechnology, Inc. (Santa Cruz, CA, USA).

### 2.3. MTT Assay

ATLL cells were seeded in 96-well plate at 2–3 × 10^4^ cells per well and PBMC were seeded 8 × 10^5^ cells per well. Cells were incubated for 24 h with varying doses of SHK (0.25–2.5 µM) in a total volume 0.1 mL per well. MTT (3-(4,5-dimethylthiazol-2-yl)-2,5-diphenyl-tetrazolium bromide (Sigma-Aldrich) was added at a final concentration of 0.5 mg/mL in each well and incubated for 3 h at 37 °C with 5% CO_2._

### 2.4. Annexin V and Propidium Iodide (PI) Staining

ED− and TL-Om1 were plated at 4 × 10^5^ cells/mL and treated with or without SHK 0.5, 1 or 2 µM for 18 h in at 37 °C with 5% CO_2_. Cells were harvested, washed by Annexin V binding buffer, and incubated with Annexin V Pacific blue (eBiosciences, San Diego, CA, USA) for 30 min in the dark at room temperature. Subsequently, propidium iodide (PI, Dojindo Laboratories) was added at a final concentration of 1 µg/mL to Annexin V-stained cells and analyzed by flow cytometry (FACSCelesta, BD Bioscience, San Jose, CA, USA). The data from flow cytometry were analyzed using FlowJo software version 10 (TreeStar, San Jose, CA, USA).

### 2.5. Cell Viability Assay

The effect of SHK on cell viability was determined using PI staining. In brief, ATLL cells were seeded at a concentration of 4 × 10^5^ cells/mL. Cells were pre-treated with or without 10 µM of Q-VD-OPh (pan-caspase inhibitor) or 5 mM of N-acetyl cysteine (NAC), a ROS scavenger for 2 h. SHK was added after 2 h of pre-treatment at a final concentration 1 µM for 18 h at 37 °C with 5% CO_2_. Cells were harvested and PI at a final concentration of 1 µg/mL was added. The cell viability was evaluated by FACSCelesta, (BD Bioscience). The data were analyzed by FlowJo software version 10 (TreeStar).

### 2.6. Intracellular ROS Measurement

Intracellular ROS was evaluated by DCFH-DA (2′-7′-dichlorodihydrofluorescein diacetate) redox-sensitive fluorescent probe (Sekisui Medical, Tokyo, Japan) staining. ED- and TL-Om1 were pre-treated with NAC for 2 h and subsequently incubated with DCFH-DA for 1 h followed by SHK treatment for 1 h at 1 μM (ED−) and 2 μM (TL-Om1). After incubation, cells were harvested for PI staining and analysed by FACSCelesta.

### 2.7. Mitochondrial Membrane Potential (Δψ) Measurement

JC-1 was used to determine the mitochondrial membrane potential. In brief, ED− or TL-Om1 cells at 4 × 10^5^ cells/mL were treated with varying doses of SHK (0, 0.5, 1, 2 μM) for 18 h. To confirm ROS upstream of mitochondrial membrane depolarization, ED− cells were pre-treated with 5 mM NAC for 2 h with or without 1 or 2 μM SHK. Cells were harvested, JC-1 stained and analyzed by flow cytometry (FACSCelesta).

### 2.8. Protein Extraction and Western Blot Analysis

Cells were treated at various concentrations with or without SHK for various times and harvested for pellet collection. Treated and untreated cells were washed in cold PBS before the addition of signal detection lysis buffer containing 25 mM HEPES (Dojindo Laboratories), 10 mM Na_4_P_2_O_7_.10H_2_O (Fujifilm Wako Pure Chemical), 100 mM NaF, 5 mM EDTA.2Na (Dojindo Laboratories), 2 mM Na_3_VO_4_ (Sigma-Aldrich), 1% Trition X-100 (Fujifilm Wako Pure Chemical) and protease inhibitor cocktail (Nacalai Tesque, Kyoto, Japan) [21]. Cells with signal detection buffer were incubated on ice for 1 h, centrifuged at 15,000 rpm for 15 min, and the supernatant of the whole cell lysate was collected. Bicinchoninic acid (BCA) protein assay (Thermo Scientific, Rockford, IL, USA) was used to determine the actual protein amount in the treated and untreated cells. Same amounts of lysed proteins (10–20 µg) were applied for multiple sodium dodecyl sulfate-polyacrylamide gel, separated by electrophoresis, and the gel was transferred by electroblotting to a polyvinylidene difluoride (PVDF) membrane (GE Healthcare Technologies, Chicago, IL, USA). The blotted membrane was blocked in non-specific binding by 5% skimmed milk in TBST washing buffer, then incubated with specific primary antibody (dilution 1:1000) and secondary horseradish peroxidase-conjugated antibody (dilution 1:2000). The proteins of interest were visualized using the ECL prime Western Blotting Detection Reagent (GE Healthcare Technologies) and analyzed using an ImageQuant LAS 4000 system (GE Healthcare Technologies). The equal amount of applied proteins was confirmed by β-action expression and equal amount of non-specific bands.

### 2.9. Xenograft Mouse Model

ED− cells (7.5 × 10^6^ cells) were subcutaneously injected into both flanks of 6- to 8-week-old male BALB/c Rag-2/Jak3 double deficient (BRJ) mice (control: 6 mice, SHK-treated: 7 mice) [22]. Three days after implantation, the mice were randomly divided into two groups. SHK was administered by oral gavage daily at 10 mg/kg twice/day for 12 days. Tumor growth was monitored by measuring maximal and minimal diameters with calipers every other day, and tumor size was estimated with the formula: tumor size mm^3^ = length (mm) × withth^2^ (mm) × 0.4, as described previously [23,24]. At the end of the experiment, tumors were removed and weighed. All mice were housed and monitored in accordance with the guidelines of the Institutional Animal Care and Use Committee of Kumamoto University.

### 2.10. Statistical Analysis

Quantitative data are shown as means ± standard deviations (SD) of three independent experiments except where specified. The statistical differences between two groups were determined by Student’s t-test or Mann–Whitney U test using SPSS statistical software (IBM, Chicago, IL, USA). *p*-values less than 0.05 were considered statistically significant.

## 3. Results

### 3.1. Antiproliferative Effects of SHK on ATLL Cells with Little Impacts on PBMC

To understand the anti-cancer effect of SHK, we first determined the cell growth inhibitory effect by MTT assay. ATLL cell lines (ED−, TL-OM1, S1T and OATL4), HTLV-1-infected cells (MT-2, MT-4) and PBMC were chosen to study the potential for anti-cancer activity. The cells were treated with 0.25–2.5 µM of SHK for 24 h. SHK appeared to suppresses the proliferation of the ATLL cell lines in a dose-dependent manner, except for MT-2 cells (Figure 1A). The half-maximal inhibitory concentration (IC_50_) of SHK for ATLL cell lines (ED−, TL-OM1, S1T and OATL4) were 0.85, 1.28, 1.13 and 1.53 μM. The IC_50_ thresholds of HTLV-1-infected cells (MT-2, MT-4) were >2.5 and 1.09 μM. The IC_50_ of SHK for PBMC was more than 2.5 μM (Figure 1B). These data indicate that SHK is an ATLL cell growth suppressor, but it did not suppress PBMC.

### 3.2. SHK Mediated Caspase-Dependent Apoptosis in ATL Cells

To confirm the role of SHK inducing cell death, the Annexin V binding assay was performed. ED− and TL-Om1 cells were selected to study the ways in which SHK induces apoptosis. The result from FACs analysis showed an increase in the Annexin V positive population with dose, suggesting that SHK induces cell apoptosis (Figure 2A). We further analyzed by Western blotting whether SHK induce apoptosis via caspase using anti-activated Caspase-8, -9 and -3. We used 1 μM for ED− and 2 μM for TL-Om1 for Western blot analysis and further experiments because SHK induced apoptosis at same level with these doses (Figure 2A). The results showed that SHK induced cleaved caspase activation in a time-dependent manner with all three indicators (Figure 2B), suggesting that SHK induces apoptosis via extrinsic and intrinsic pathways. To examine whether these caspases are involved in SHK-induced apoptosis, the pan-caspase inhibitor Q-VD-OPh was used. In particular, combined treatment of SHK and Q-VD-OPh diminished SHK-induced cell death (Figure 2C). Taken together, SHK induces caspase-dependent apoptosis via extrinsic and intrinsic pathways.

### 3.3. Accumulation of Intracellular ROS Involved in SHK-Induced ATLL Cell Death

A previous report [25] showed that generation of ROS is associated with apoptosis. To identify the role of SHK inducing cell death via intracellular ROS in ATLL cells, the ROS detector DCFH-DA was monitored. Intracellular ROS oxidize non-fluorescing DCFH to DCF probes, which can be detected by flow cytometry. Moreover, N-acetyl-L-cysteine (NAC), a ROS scavenger, was used as a co-treatment. SHK exhibited production of ROS in both ED− and TL-Om1 cells. The ROS generation was diminished by the combination with NAC (Figure 3A). We next investigated the ways in which ROS production induced by SHK is associated with cell death. ED− and TL-Om1 co-treatment with SHK and NAC clearly reversed the effect of SHK-induced cell death (Figure 3B), implying that SHK induces cell death via ROS activation.

### 3.4. ROS Are Upstream of SHK Mediated Mitochondria Depolarization

Accumulation of ROS causes depolarization of the mitochondrial membrane potential (ΔΨm). A previous report [17] showed that SHK accelerates ROS accumulation, disrupting the mitochondrial membrane and inducing apoptosis. We examined the ΔΨm in ED− cells and TL-Om1 cells after SHK treatment using JC-1 dye, a lipophilic cationic carbocyanine dye that accumulates in mitochondria. Red fluorescence represented J-aggregates indicating a high ΔΨm while green fluorescence represented J-monomer, a low ΔΨm. Our results indicated that SHK treatment increased the percentage of cells with low ΔΨm in a dose-dependent manner: 0–2 μM-yielded 1.32, 2.69, 12.00 and 17.40% (Figure 4A,B). In addition, we confirmed ROS to be upstream of mitochondria depolarization using ROS scavenger NAC. Co-treatment with NAC abrogated the percentage of cells with low ΔΨm (Figure 4C). SHK also modulated the vital protein of mitochondria expression (Figure 4D). Cytochrome c, a protein of the inner membrane of mitochondria which is released to cytosol upon apoptosis induction, increased in a time-dependent manner after SHK treatment (ED−: 1 μM, TL-Om1: 2 μM) while pro-survival proteins, p-Bcl2 and survivin, were decreased after SHK treatment in ED− cells, although p-Bcl2 was not detected in TL-Om1 cells. Taken together, these results indicate that ROS are upstream of mitochondrial damage induced by SHK.

### 3.5. ROS Induced by SHK Activates ER Stress and Induces Apoptosis

A recent study reported that SHK induced ROS generation, and that activation of ER stress leads to mitochondrial apoptosis in cancer cells [26]. Thus, ER stress protein markers were screened in ED− and TL-Om1 cells. Western blot analysis revealed that SHK quickly accelerated the expression of ER stress proteins from 1 to 6 h, including p-eIF2α, ATF4, XBP-1, p-JNK, and finally CHOP (Figure 5A). These results led us to further examine the role of ROS in ER stress. To identify whether or not ROS are an upstream of ER stress, SHK was co-treated with the ROS scavenger NAC. Combination treatment rescued the expression of ER stress-related proteins induced by SHK. To confirm ROS-mediated ER stress induces apoptosis, ED− cells were co-treated with SHK and ISRIB, an integrated stress response inhibitor. As shown in Figure 5C,D, ISRIB partially inhibited SHK-induced apoptosis. Overall, these findings suggested that ROS are upstream of the ER stress induced by SHK, and ER stress induced apoptosis for ATLL cells.

### 3.6. SHK Suppressed ATLL Cell Growth in an ATLL-Xenografted Mouse Model

We investigated the anti-ATLL effect of SHK in vivo. ED− cells (7.5 × 10^6^ cells) were subcutaneously inoculated into both flanks of 6–8-week-old male BRJ mice (control: 6 mice, treated: 7 mice), and SHK was administered by oral gavage daily at 10 mg/kg twice/day for 12 days starting from 3 days after inoculation. As shown in Figure 6A,B, SHK treatment significantly suppressed tumor growth (control: 666.27 ± 173.40 mm^3^, treated: 353.93 ± 144.49 mm^3^, *p* < 0.01, on day 15). When the mice were sacrificed at day 15, the tumor weight in SHK-treated mice was significantly smaller than that of controls group (control: 1.03 ± 0.31 g, treated: 0.68 ± 0.30 g, *p* < 0.01,) (Figure 6C). The SHK-treated mice appeared to be healthy and their body weight was the same as that of the untreated mice at the time of sacrifice (Figure 6D). These results suggest that oral administration of SHK inhibits ATLL growth in immunodeficient mice without adverse effects.

## 4. Discussion

The clinical course of ATLL is aggressive and generally refractory to conventional chemotherapy, and ATLL develops in the elderly. Novel therapeutic strategies with fewer side effects are needed. We investigated the effects of the Shikonin, a naphthalene ingredient, on ATLL cells in vitro and in vivo. SHK induced apoptosis of ATLL cells via induction of ROS and ER stress, but did not affect peripheral blood mononuclear cells. We also determined that SHK suppressed tumor growth in ATLL-inoculated mice without apparent adverse effects, suggesting SHK to be a candidate for anti-ATLL therapy via a unique mechanism.

Various biological activities have been reported for SHK, including anti-inflammation, anti-cancer, cardiovascular protection, and anti-microbiome effects, mainly by regulating the NF-κB, PI3K/Akt/MAPKs, Akt/mTOR, TGF-β, GSK3β, TLR4/Akt signaling pathways, i.e., the main pathways of cancer cell survival [14]. These inhibitory effects on cancer involve multiple mechanisms related to the inhibition of cell proliferation and induction of apoptosis [27,28,29], inhibition of migration and invasion [30], cell cycle arrest [31], suppression of aerobic glycolysis [32], and overcoming drug resistance [33]. We showed here that SHK rapidly induced apoptosis of ATLL cells by accumulation of ROS followed by mitochondria damage (Figure 3 and Figure 4) and ER stress (Figure 5). The fact that pre-treatment with NAC suppressed not only ROS induction but also mitochondria damage and ER stress in ATLL cells indicates that the induction and accumulation of ROS are primary events for inducing apoptosis against ATLL cells.

The ER and mitochondria play crucial roles in regulating cellular homeostasis and cell death, especially cross-talk of these organelles mediated by Calcium ion are important for survival of cancer cells [34]. The link between induction of ROS, activation of the ER stress pathway, and apoptotic death has been demonstrated in various cancer cell lines by various natural compounds [35,36,37,38]. We showed here that SHK rapidly induced apoptosis of ATLL cells by activating both the extrinsic or death receptor pathway and the intrinsic or mitochondrial pathway (Figure 2). SHK is a ROS inducer, and accumulation of ROS in cancer cells plays an important role upstream of the apoptosis pathway [39]. SHK induces ROS-related apoptotic cell death in human prostate cancer [26] and primary effusion lymphoma [17]. We showed that ROS accumulated in ATLL cells with SHK treatment and induced cell death. This fact was attenuated by pre-treatment by an antioxidant, NAC, indicating that generation of ROS is the initiating event of SHK-induced apoptosis of ATLL cells (Figure 7). In addition, the fact that ISRIB partially inhibited SHK-induced apoptosis indicates that ER stress may be related to mitochondria dysfunction and activation of apoptosis pathway. SHK treatment induced apoptosis of ATLL cells but not of PBMC or a HTLV-1 infected cell line (MT-2). In addition, the SHK-treated mice remained healthy during treatment, suggesting that SHK selectively induces ROS production and accumulation in ATLL cells. Since SHK also does not lead to drug resistance [33], SHK is an attractive candidate for combination therapy against ATLL.

## 5. Conclusions

The present study reported that SHK inhibited proliferation of ATLL cells in vitro by causing apoptosis through induction of ROS, mitochondria depolarization and ER stress. SHK also inhibited the growth of ATLL tumors in xenografted mice. It is a promising potential reagent for ATLL therapy.

## Figures and Tables

**Figure 1 antioxidants-12-00864-f001:**
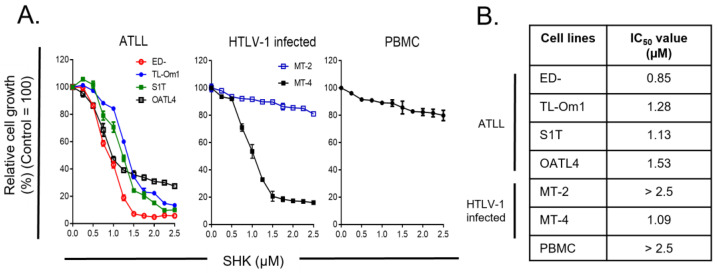
Anti-proliferative effect of SHK on ATLL. (**A**) ATLL, HTLV-1-infected cells and PBMC were treated with or without SHK for 24 h, and cell proliferation ability was measured by MTT assay (mean ± SE). (**B**) IC50 values were analyzed using GraphPad software.

**Figure 2 antioxidants-12-00864-f002:**
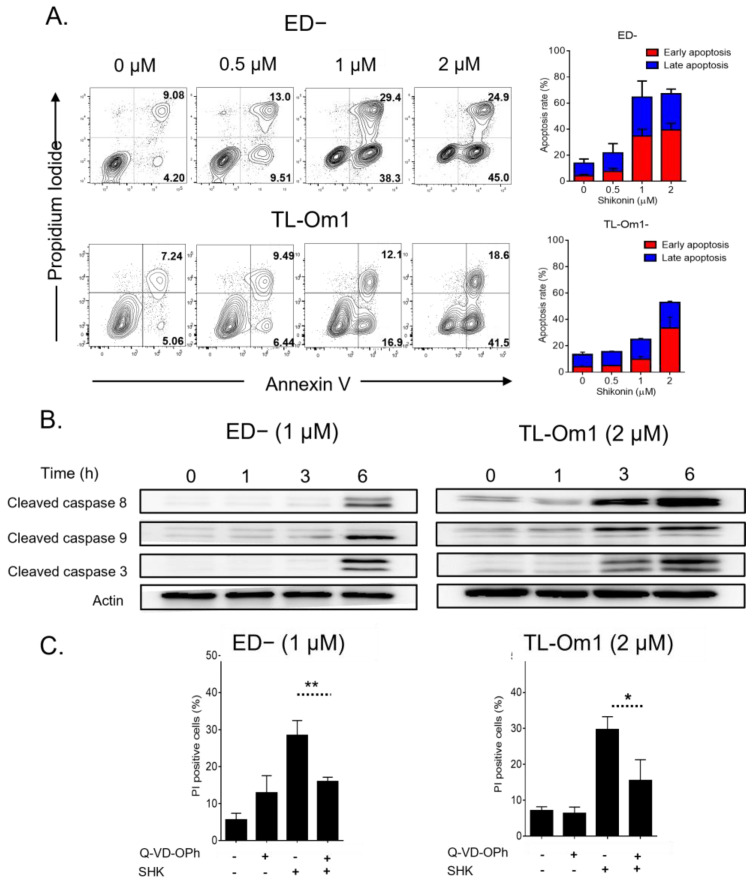
Induction of caspase-dependent apoptosis in ATLL cells by SHK. (**A**) Human ED− and TL-Om1 cells were treated with/without SHK for 18 h (0, 0.5, 1 and 2 μM). Cells were stained with annexin V and PI. Apoptosis was measured using flow cytometry. (**B**) Human ED− and TL-Om1 cells were treated with or without SHK for 18 h (1 and 2 μM), respectively. Apoptosis markers were analyzed by Western blotting. (**C**) Human ED− and TL-Om1 cells were pre-treated with Q-VD-OPh in a concentration of 20 μM for 2 h, and then with or without SHK (1 and 2 μM) for 18 h. Cell death was measured by PI staining and flow cytometry. * *p* < 0.05, ** *p* < 0.01.

**Figure 3 antioxidants-12-00864-f003:**
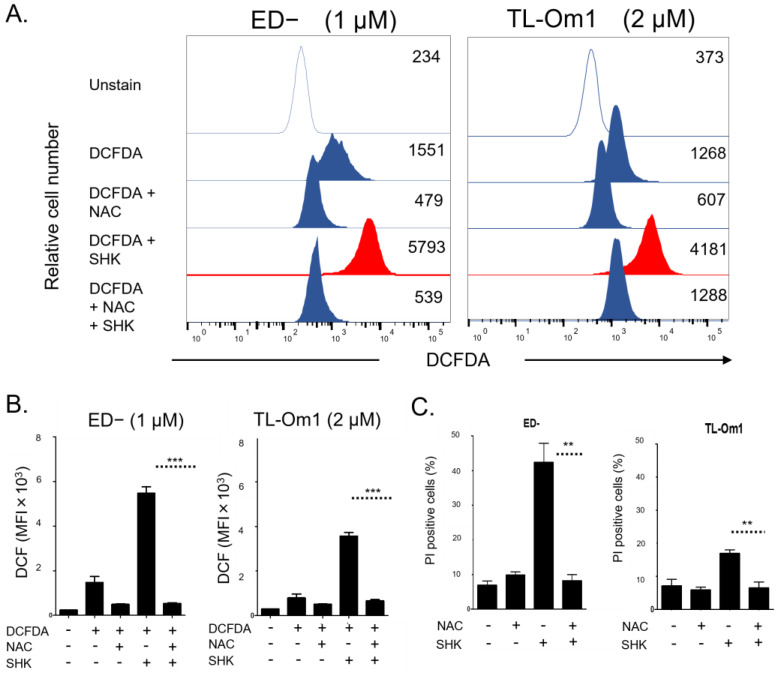
Accumulation of ROS and induction of cell death in ATLL cells. (**A**,**B**) Human ED− and TL-Om1 cells were pre-treated with/without NAC for 2 h, DCFDA (10 μM) was added for 1 h followed by treatment with or without SHK at 1 and 2 μM for ED− and TL-Om1, respectively. ROS levels were measured by flow cytometry. (**C**) Human ED− and TL-Om1 cells were pre-treated with or without NAC for 2 h, and subsequently treated with or without SHK at 1 and 2 μM for ED− and TL-Om1, respectively. Cell death was determined using PI staining and flow cytometry. ** *p* < 0.01, *** *p* < 0.001.

**Figure 4 antioxidants-12-00864-f004:**
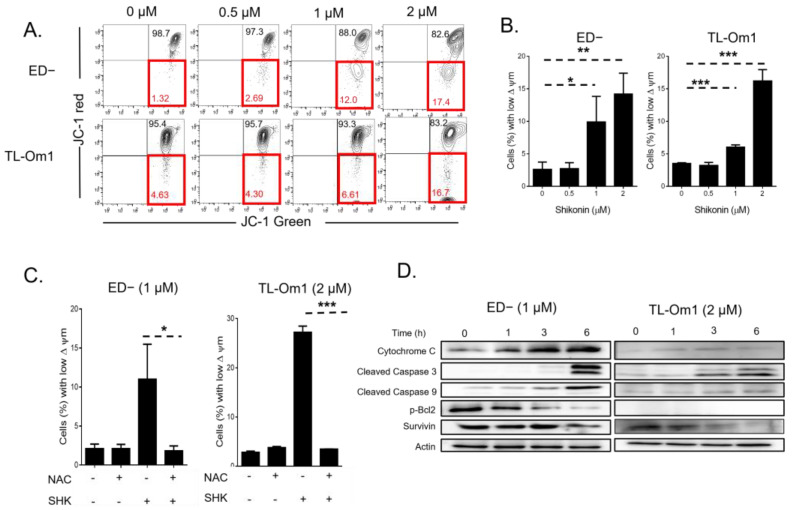
Induction of mitochondria depolarization through ROS with SHK treatment. (**A**) ED− and TL-Om1 cells were treated with/without SHK for 18 h (0, 0.5, 1 and 2 μM). For measurement of Δψ, cells were stained with fluorochrome dye JC-1 and analyzed by flow cytometry. Dead cells were excluded by PI staining. Green fluorescent positive cells indicated disruption of Δψ. (**B**) Bar graph represents percentage of cells with low Δψ. (**C**) ED− and TL-Om1 cells were pre-treated with or without NAC for 2 h, and subsequently treated with or without SHK (ED−: 1 μM, TL-Om1: 2 μM) for 18 h. Cells were stained by PI and measured by flow cytometry. (**D**) ED− and TL-Om1 cells were treated with/without SHK (ED−: 1 μM, TL-Om1: 2 μM) for 0, 1, 3 or 6 h. Mitochondria depolarization and apoptosis markers were determined using Western blotting. * *p* < 0.05, ** *p* < 0.01, *** *p* < 0.001.

**Figure 5 antioxidants-12-00864-f005:**
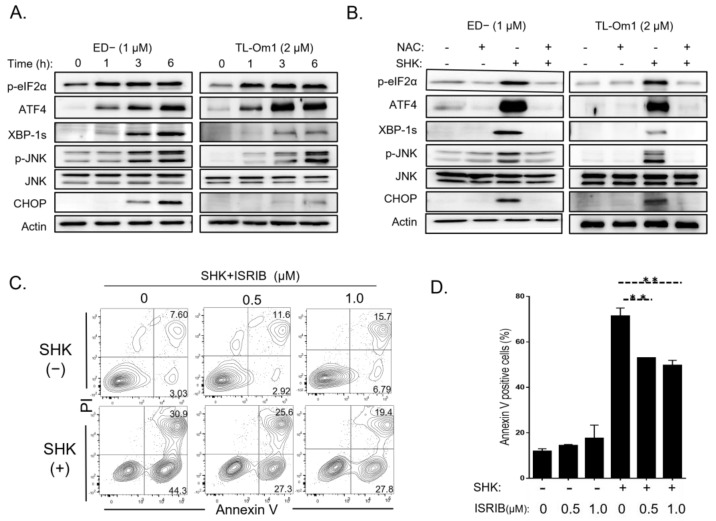
Induction of ER stress in ATLL cells through accumulation of ROS. (**A**) ED− and TL-Om1 cells treated with or without SHK for 0–6 h with 1 and 2 μM for ED− and TL-Om1, respectively. Activation of ER stress markers was determined by Western blotting. (**B**) Human ED− and TL-Om1 cells were pre-treated with/without NAC for 2 h, and then treated with or without SHK for 6 h. Activation of ER stress markers was determined by Western blotting. (**C**,**D**). ED− cells were pre-treated with ISRIB for 1 hr and treated with 1 μM of SHK for 12 h. Cells were harvested and stained with Annex-n V and PI, and analyzed with flow cytometry. ** *p* < 0.01.

**Figure 6 antioxidants-12-00864-f006:**
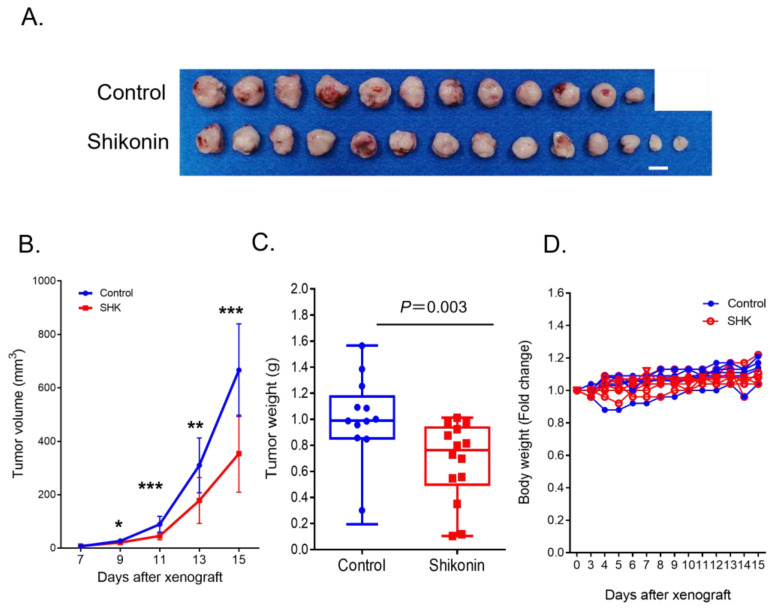
SHK inhibited ATLL cell growth in vivo. ED− cells were subcutaneously injected into both flanks of BALB/c RJ mice. SHK 10 mg/kg was administered by oral gavage twice daily for 15 consecutive days. Tumor was allowed to develop for three days before starting treatment. Control mice were injected with DMSO. (**A**,**B**) Tumor growth was monitored by measuring maximal and minimal diameters with calipers every other day, and tumor size was estimated with the formula: tumor size mm^3^ = length (mm) × width^2^ (mm) × 0.4, as described previously [23,24]. * *p* < 0.05, ** *p* < 0.01, *** *p* < 0.001. (**C**) Tumor weight was measured at the end of the experiment. Data are shown as box-and whisker plot. Each dot represents the weight of a single mass. (**D**) Body weight was measured daily.

**Figure 7 antioxidants-12-00864-f007:**
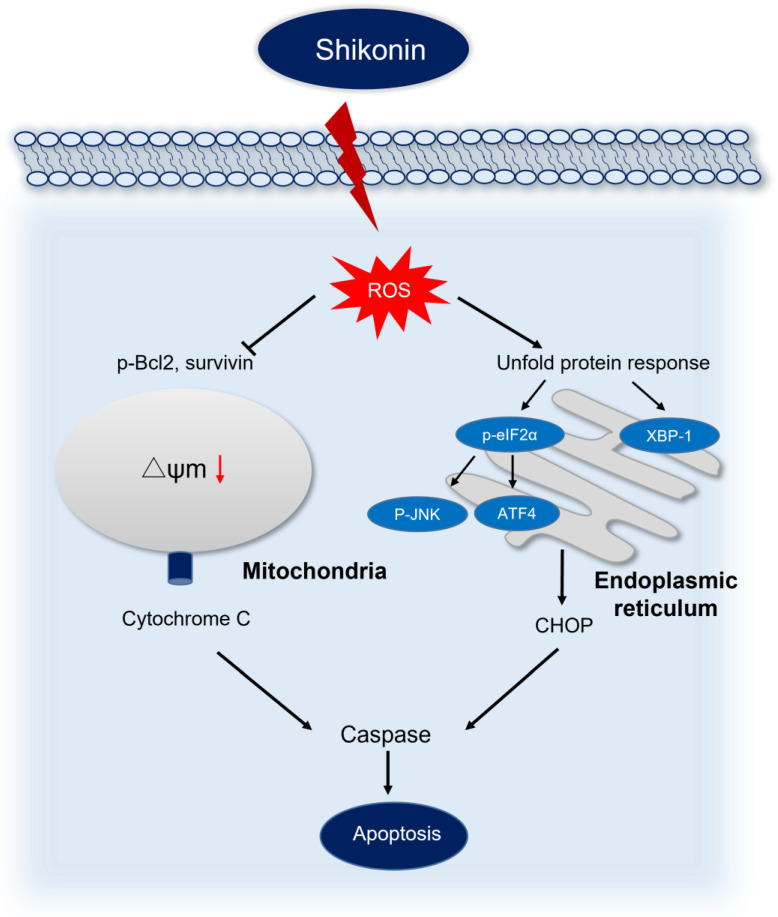
The proposed model of SHK-induced apoptosis. SHK-induced ROS accumulation causes mitochondria depolarization and release of cytochrome c. Separately, ROS activates ER stress markers leading to apoptosis.

## Data Availability

The datasets used and/or analyzed during current study are available from the corresponding author on reasonable request. Data is contained within the article.

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
