# Peer review of "Shikonin Induces ROS-Dependent Apoptosis Via Mitochondria Depolarization and ER Stress in Adult T Cell Leukemia/Lymphoma"

_antioxidants, 2023, doi:10.3390/antiox12040864_

Round 1

Reviewer 1 Report

In this study, the authors elucidate the effect of Shikonin as an important inducer to ROS production and accumulation in ATLL cells.

The study design was well conceived and approached with different and valid techniques. However, there are minor revisions to be made. I wish the authors good luck in the publication process.

Minor revision:

Figure 2: In the description of panel B, the authors refer to a treatment time of 18h and to two different concentrations of SHK. In reality, the western blotting images describe a single concentration and different times. Edit the caption of the figure and clarify which concentration of SHK was used. Similarly, in panel C clarify which concentration of SHK was used.

Figure 3: Replace SOS with ROS and clarify which concentration of SHK was used.

Section 3.4: The data shown are very interesting but concern only one cell line, the ED-. To strengthen your concept, it is necessary to show the same analyses also on the TL-Om.

Figure 5: Clarify the concentration of SHK used

Careful reading is recommended in order to eliminate the various typing errors

Author Response

Reviewer 1

Comments and Suggestions for Authors

In this study, the authors elucidate the effect of Shikonin as an important inducer to ROS production and accumulation in ATLL cells.

The study design was well conceived and approached with different and valid techniques. However, there are minor revisions to be made. I wish the authors good luck in the publication process.

 Minor revision:

Figure 2: In the description of panel B, the authors refer to a treatment time of 18h and to two different concentrations of SHK. In reality, the western blotting images describe a single concentration and different times. Edit the caption of the figure and clarify which concentration of SHK was used. Similarly, in panel C clarify which concentration of SHK was used.

【Authors】 Thank you very much for your kind suggestions. The concentration that we use were 1 and 2µM for ED- and TL-Om1 respectively, and treated in different time point. We added the concentration of SHK in all of Figures. We selected 1 for ED- and 2µM forTL-Om1, because these doses are almost equivalent to induce apoptosis in Figure 2A. We also added explanation at Line 174-175.

Figure 3: Replace SOS with ROS and clarify which concentration of SHK was used.

【Authors】 Thank you very much for your kind suggestions. We corrected typo “SOS” to “ROS” on the title of Figure 3. The concentration of SHK was 1μM for ED-  and 2μM for TL-Om1, respectively , and it was on the Figure legends. We also added the used dose on Figure 2-5.

Section 3.4: The data shown are very interesting but concern only one cell line, the ED-. To strengthen your concept, it is necessary to show the same analyses also on the TL-Om.

【Authors】 Thank you very much for your kind suggestions. According to the reviewer’s suggestion, we added the data of TL-Om1 in Figure 4. As dead cells also express green fluorescence which cannot delete from fluorescence microscopy photos, we deleted microscope photos. (In case of Flowcytometory, we could exclude dead cells by PI staining).

Figure 5: Clarify the concentration of SHK used

【Authors】Thank you very much for your kind suggestions. For the concentration that we used in this experiment were 1 and 2µM for ED- and TL-Om1 respectively. We added the concentration in Figure 5.

Careful reading is recommended in order to eliminate the various typing errors

【Authors】Thank you very much for your kind suggestions. We carefully checked the text of our manuscript and corrected the typing errors.

Reviewer 2 Report

In their manuscript entitled “Shikonin induces ROS dependent apoptosis via mitochondria depolarization and ER stress in Adult T Cell Leukemia/Lymphoma”, Okada and colleagues evaluated the anticancer efficacy of the phytochemical compound Shikonin (SHK) on adult T-cell leukemia/lymphoma (ATLL) in vitro and in vivo. They found that SHK induces the accumulation of ROS, which in turn drive mitochondrial depolarization, ER stress, and apoptosis in ATLL cells but not in healthy PBMCs. The manuscript is clearly written and the results are interesting. However, before publication the manuscript needs a couple of revisions.

The most intriguing observation is the cleavage of caspase 8 following SHK treatment. The Authors correctly concluded that this indicates an activation of extrinsic apoptosis. However, as the name states, extrinsic apoptosis requires extrinsic stimuli, which signal to the cell committed to die via a direct interaction between ligands and their receptors. It is very interesting that an internal stimulus, such as Shikonin, activates the extrinsic apoptotic pathway. Could the Authors hypothesize how this happens?

Figure 4D: as the Authors wrote, cytochrome c is released from mitochondria during apoptosis. Its total expression does not increase. So, I guess this western blot, showing a dose-dependent increase of cytochrome c expression, should represent the cytosolic fraction of cells. However, in the Materials and Methods section it seems that the Authors used a lysis buffer that disrupt all cell compartments. Thus, why is cytochrome c increasing?

Line 236: which form of XBP1? The unspliced form per se does not indicate ER stress.

Figure 6A: it seems that two tumors in the control group were covered.

Figure 7: the Authors do not have any evidence to claim that there is a causal link between ER stress and apoptosis. Data regarding the role of ROS are convincing because they used NAC, but they never used a pharmacological inhibitor of ER stress or a genetic approach to modulate CHOP expression. I suggest to do it. If they think that CHOP is regulated by the phospho-eIF2a-ATF4 axis, they could try with ISRIB. Otherwise, they cannot claim that caspase cleavage is induced by CHOP, ruling out that ER stress could be simply a ROS-induced epiphenomenon, or even a counteractive circuitry.

Other points:

Line 161: I understood what the Authors meant, but this sentence is misleading. It seems that only caspase 3 is a marker of apoptosis, as extrinsic and intrinsic pathways regulated by caspase 8 and 9 would not be apoptotic. For the sake of comprehension, it is sufficient to mention only extrinsic and intrinsic apoptotic pathways, without specifying that caspase 3 is an effector. 

Line 186: the semicolon should be removed. 

Line 231: this title is confused.

Line 256: 18 consecutive days? In the text the Authors wrote 12, indeed the maximum value in the x-axis is 15. 

Lines 258-260: this caption is confused. (C) is not tumor volume, and I don't understand (D) at the end of line 258. Obviously, it is not possible to measure tumor mass (as shown in A) every three days. Moreover, according to the graph in B, the tumor volume was measured every 2 days.

Line 275: there is a typo (ani-microbiome).

Line 281: "apoptosis of ATLL cells".

Author Response

Reviewer 2

Comments and Suggestions for Authors

In their manuscript entitled “Shikonin induces ROS dependent apoptosis via mitochondria depolarization and ER stress in Adult T Cell Leukemia/Lymphoma”, Okada and colleagues evaluated the anticancer efficacy of the phytochemical compound Shikonin (SHK) on adult T-cell leukemia/lymphoma (ATLL) in vitro and in vivo. They found that SHK induces the accumulation of ROS, which in turn drive mitochondrial depolarization, ER stress, and apoptosis in ATLL cells but not in healthy PBMCs. The manuscript is clearly written and the results are interesting. However, before publication the manuscript needs a couple of revisions.

The most intriguing observation is the cleavage of caspase 8 following SHK treatment. The Authors correctly concluded that this indicates an activation of extrinsic apoptosis. However, as the name states, extrinsic apoptosis requires extrinsic stimuli, which signal to the cell committed to die via a direct interaction between ligands and their receptors. It is very interesting that an internal stimulus, such as Shikonin, activates the extrinsic apoptotic pathway. Could the Authors hypothesize how this happens?

【Authors】Thank you very much for your kind suggestions. From our result showed that SHK induced ER stress and apoptosis. Previous study has been reported that after prolong of ER stress, IRE1 signalling is attenuated, and the PERK pathway ultimately leads to apoptosis (Lu, Lawrence et al. 2014). During the terminal phase of the UPR, RNA polymerase II-associated protein 2 (RPAP2) phosphatase acts downstream of PERK to attenuate IRE1 signalling allowing TRAILR2/DR5 mRNA translation and the execution of apoptosis via the extrinsic pathway (Chang, Lawrence et al. 2018). Therefore, we hypothesized that SHK induced ER stress might induced extrinsic pathway via activation of DR5 (Mora-Molina and Lopez-Rivas 2022).

Reference:

Chang, T. K., D. A. Lawrence, M. Lu, J. Tan, J. M. Harnoss, S. A. Marsters, P. Liu, W. Sandoval, S. E. Martin and A. Ashkenazi (2018). "Coordination between Two Branches of the Unfolded Protein Response Determines Apoptotic Cell Fate." Mol Cell 71(4): 629-636 e625.

Lu, M., D. A. Lawrence, S. Marsters, D. Acosta-Alvear, P. Kimmig, A. S. Mendez, A. W. Paton, J. C. Paton, P. Walter and A. Ashkenazi (2014). "Opposing unfolded-protein-response signals converge on death receptor 5 to control apoptosis." Science 345(6192): 98-101.

Mora-Molina, R. and A. Lopez-Rivas (2022). "Restoring TRAILR2/DR5-Mediated Activation of Apoptosis upon Endoplasmic Reticulum Stress as a Therapeutic Strategy in Cancer." Int J Mol Sci 23(16).

Figure 4D: as the Authors wrote, cytochrome c is released from mitochondria during apoptosis. Its total expression does not increase. So, I guess this western blot, showing a dose-dependent increase of cytochrome c expression, should represent the cytosolic fraction of cells. However, in the Materials and Methods section it seems that the Authors used a lysis buffer that disrupt all cell compartments. Thus, why is cytochrome c increasing?

【Authors】 The authors thank the reviewer for the kind concern. The lysis buffer we used can lyse cytoplasmic and nuclear protein, but not much lyse mitochondria protein (Chaiyarit, S. and V. Thongboonkerd. "Comparative analyses of cell disruption methods for mitochondrial isolation in high-throughput proteomics study." Anal Biochem 394(2): 249-258. 2009). So we could detect mostly leaked cytochrome c in our western blot. We added more information of our lysis buffer in materials and methods (Lines 113-120)

Line 236: which form of XBP1? The unspliced form per se does not indicate ER stress.

【Authors】 We used rabbit anti-XBP1 (M-186) (SC-7160, Santa Cruz), which detects both active isoform (XBP-1S: 28kDa) and inactive isoform (XBP-1U: 32KDa) (.https://datasheets.scbt.com/sc-7160.pdf). XBP-1S is an active transcription factor that plays a vital role in the unfolded protein response (UPR). As the reviewer’s suggestion, the XBP-1 unspliced form does not indicate ER stress.  Unspliced Xbp1 mRNA is cleaved by the activated stress sensor IRE1α and converted to the mature form encoding spliced XBP1 (XBP1s). (https://pubmed.ncbi.nlm.nih.gov/34356855/). We found XBP-1S is upregulated with SHK treatment as shown in Figure 5., indicating that ER stress occurred with AHK treatment. We changed name from “XBP-1” to XBP-1S) on Figure 5.

Figure 6A: it seems that two tumors in the control group were covered.

【Authors】  The authors thank the reviewer for the kind concern. One of the control mouse transplanted, was getting sick and rapidly reduced body weight with unknown reason. So we excluded this mice for analysis.

Figure 7: the Authors do not have any evidence to claim that there is a causal link between ER stress and apoptosis. Data regarding the role of ROS are convincing because they used NAC, but they never used a pharmacological inhibitor of ER stress or a genetic approach to modulate CHOP expression. I suggest to do it. If they think that CHOP is regulated by the phospho-eIF2a-ATF4 axis, they could try with ISRIB. Otherwise, they cannot claim that caspase cleavage is induced by CHOP, ruling out that ER stress could be simply a ROS-induced epiphenomenon, or even a counteractive circuitry.

【Authors】 The authors thank the reviewer for the kind concern. Regarding the casual link between ER stress and apoptosis, we performed additional experiments used by ISRIB, an integrated stress response inhibitor, that reverses the effects of eIF2α phosphorylation, and fond that ISRIB partially inhibited SHK-induced apoptosis, indicating that ER stress activated apoptosis pathway. We added these data at Figure C and D, and discussion a Line 314-328.

Other points:

Line 161: I understood what the Authors meant, but this sentence is misleading. It seems that only caspase 3 is a marker of apoptosis, as extrinsic and intrinsic pathways regulated by caspase 8 and 9 would not be apoptotic. For the sake of comprehension, it is sufficient to mention only extrinsic and intrinsic apoptotic pathways, without specifying that caspase 3 is an effector. 

【Authors】Thank you very much for your comments. We deleted these words.

Line 186: the semicolon should be removed. 

【Authors】 The authors thank the reviewer for the kind suggestion. We already removed as reviewer suggestion.

Line 231: this title is confused.

【Authors】Would it be better if we change to ‘ ROS is an upstream of SHK induce ER stress’  ROS induced by SHK activates ER stress.

Line 256: 18 consecutive days? In the text the Authors wrote 12, indeed the maximum value in the x-axis is 15. 

【Authors】 The authors thank the reviewer. It is our mistake and we already edited to 15 days

Lines 258-260: this caption is confused. (C) is not tumor volume, and I don't understand (D) at the end of line 258. Obviously, it is not possible to measure tumor mass (as shown in A) every three days. Moreover, according to the graph in B, the tumor volume was measured every 2 days.

【Authors】We are sorry for our mistake. We corrected the Figure legends.

Line 275: there is a typo (ani-microbiome).

【Authors】Thank you very much. We corrected the word “ani-microbiome” to “anti-microbiome”

Line 281: "apoptosis of ATLL cells".

【Authors】 The authors thank the reviewer, we already edited